# Correlation between PlateletWorks^®^ and PFA-100^®^ for Measuring Platelet Function before Urgent Surgery in Patients with Chronic Antiplatelet Therapy

**DOI:** 10.3390/jcm10020255

**Published:** 2021-01-12

**Authors:** Rafael Anaya, Mireia Rodriguez, José María Gil, Noelia Vilalta, Angela Merchan-Galvis, Victoria Moral, José Mateo, María José Martinez-Zapata

**Affiliations:** 1Anesthesiology Service, Hospital de la Santa Creu i Sant Pau, 08041 Barcelona, Spain; rafaelanaya30@yahoo.es (R.A.); mrodriguezpr@santpau.cat (M.R.); jgils@santpau.cat (J.M.G.); nvilalta@santpau.cat (N.V.); vmoralg@santpau.cat (V.M.); 2Iberoamerican Cochrane Centre-Public Health and Clinical Epidemiology, IIBSant Pau, 08025 Barcelona, Spain; amerchang@santpau.cat; 3Departamento de Medicina Social y Salud Familiar, Universidad del Cauca, 190003 Popayán, Colombia; 4Thrombosis and Hemostasis Unit, Hematology Department, Hospital de la Santa Creu i Sant Pau, 08041 Barcelona, Spain; jmateo@santpau.cat; 5Centro de Investigación Biomédica en Red de Epidemiología y Salud Pública (CIBERESP), 08025 Barcelona, Spain

**Keywords:** Platelet Function Test, point-of-care system, antiplatelet therapy, drug monitoring, urgent surgery

## Abstract

Hemostasis is crucial for reducing bleeding during surgical procedures. The points-of-care based on the platelet function test could be useful to minimize the complications related to chronic antiplatelet therapy during surgery. The present study is aimed at comparing two point-of-care platelet function devices—Platelet Function Analyzer PFA-100^®^ (Siemens Canada, Mississauga, ON, Canada) and Plateletworks^®^(Helena Laboratories, Beaumont, TX, USA). Our objective is to evaluate if they provide comparable and useful information to manage anti-aggregate patients before surgery. We included patients with a femoral fracture receiving chronic antiplatelet therapy and a median age of 89 years (range from 70 to 98). A platelet function evaluation was performed on all patients before surgery using both devices—Plateletworks^®^ and PFA-100^®^. The correlation between Plateletworks^®^ and PFA-100^®^ was performed using Cohen’s Kappa coefficient. Twenty consecutive patients participated in the trial; 16 patients were under treatment with 75 mg/day of clopidogrel, three with >300 mg/day of acetylsalicylic acid (ASA), and only one was in treatment with both antiplatelet agents. Cohen’s Kappa coefficient was 0.327 comparing PFA-100^®^-ADP (adenosine diphosphate) and Plateletworks^®^ and, 0.200 comparing PFA-100^®^-EPI (epinephrine) and Plateletworks^®^. In conclusion, we found a weak concordance comparing PFA-100^®^ and Plateletworks^®^. This could partially be due to the advanced age of the included patients. However, given the limited sample size, more studies are necessary to confirm these results.

## 1. Introduction

In surgical procedures, hemostasis is crucial for reducing bleeding and wound healing. However, it can be altered for different causes, such as platelet stimulation, deficiency of receptors for fibrinogen (Gp II_b_ III_a_) and von Willebrand factor (Gp I_b_), and decreased platelet aggregation secondary to administration of heparin, as well as the presence of platelet dilution [1]. Added to this, the widespread use of antiplatelet medication and the high proportion of patients undergoing surgery with chronic antiplatelet drug prescriptions makes it more difficult to reach balanced hemostasis.

During the last 30 years, the analysis of the platelet function close to the patient (point-of-care testing) has been used in surgery, and it is associated with a decline in transfusion and bleeding-related outcomes. Previous studies assessed the efficiency of blood coagulation using ThromboElastoGraphy (TEG) in the framework of a transfusion procedure and showed a decrease in blood transfusion [2]. On the other hand, it has been demonstrated that the use of antiplatelet therapy has been associated with postoperative bleeding [3]. For this reason, the preoperative quantification of platelet function is proposed in a few guidelines [4] to know whether the patients presenting for surgery have platelet inhibition or not.

Traditionally, platelet function has been measured using light transmission aggregometry (LTA). This method, currently, diagnoses a variety of acquired and congenital platelet disorders. Nevertheless, the test requires an important quantity of blood, qualified laboratory technicians, and expert personnel to explain the results [5,6]. These inconveniences limit its use during current practice. For these reasons, there are numerous point-of-care devices marketed for measuring platelet function, which has been evaluated in both cardiac and non-cardiac surgery [1].

The present study is aimed at comparing two point-of-care platelet function devices—Platelet Function Analyzer PFA-100^®^ (Siemens Canada, Mississauga, ON, Canada) and Plateletworks^®^ (Helena Laboratories, Beaumont, TX, USA). Both are used to study platelet function according to their capacity to aggregate. The evidence regarding the use of these devices in surgery is scarce. However, both devices are cheap, easy to use, and affordable for most centers. Our objective is to evaluate if they provide comparable and useful information to manage anti-aggregate patients before surgery to avoid complications and optimize surgery timings. 

## 2. Materials and Methods

The protocol of this study has been published previously [7]. The main study is an ongoing, multicenter, randomized, open-label, parallel clinical trial. The principal outcome of this trial is to assess if the strategy of measuring preoperative platelet function in patients with chronic antiplatelet drugs can reduce the time from hospital admission to operation under spinal anesthesia and whether it affords any benefits in comparison with the usual approach centered on postponing surgery and stopping antiplatelet drug therapy.

This paper corresponds to the results of the sub-study that assesses the concordance between PFA-100^®^ and Plateletworks^®^ in a subgroup of 20 patients at one of the participant’s sites.

The clinical trial accomplished the ethical principles of the Declaration of Helsinki. The Institutional Review Board of the participant site approved the study protocol. Before inclusion, all patients received information about the study and signed the informed consent. The study protocol was registered at clinicals.gov with the code number NCT03231787.

### 2.1. Patients

We included patients with chronic antiplatelet drugs admitted to the trauma urgency section for a femoral fracture. All study participants were randomized to premature surgery with spinal anesthesia (experimental group) if the number of functional platelets measured by Plateletworks^®^ was superior to 80 × 10^9^/L or delayed surgery (control group) also with spinal anesthesia. Before surgery, all patients underwent at least a platelet function testing.

For this sub-study, we enrolled in the first consecutive 20 patients who agreed to participate between September 2017 and June 2018 at one of the participating hospitals. 

### 2.2. Platelet Function Tests

A platelet function evaluation was performed on all patients before surgery using both devices—Plateletworks^®^ and PFA-100^®^. 

Plateletworks^®^ is an in vitro analytic screening test that quantifies the percentage of aggregated platelets or the percentage of inhibited. In our trial, blood samples were collected by venipuncture. The first milliliter of blood was drowned in an approach to avoid contamination with tissue thromboplastin. Within one minute after completing the draw, the 3 mL of blood was added to the ethylenediaminetetraacetic acid (EDTA) and 3 mL to agonists (ADP or arachidonic acid) tubes. The samples were carried to the laboratory to perform on the platelet count and to record it. Testing was performed in less than 10 min after sampling. 

The Plateletworks^®^ involves two steps: using a cell counter to determine total platelets in a sample of blood with EDTA and subsequently verifying the functional platelet count on a second blood sample exposed to a well-known platelet agonist like ADP (used in patients treated with clopidogrel) or the arachidonic acid (for patients receiving salicylic acid). These substances activate those platelets that are functional; in consequence, platelets will constitute clumps, and they will not be counted by the cell counter. The difference in the platelet count between samples first and second provides a direct measurement of the functional platelet number. We considered that patients presenting a count of over 80 × 10^9^/L of functioning platelets were suitable for surgery under spinal anaesthesia [8,9]. 

The PFA-100^®^ mimics a vessel wall under physiologic stressful conditions. It comprises a membrane coated with collagen and ADP (PFA-100^®^-ADP) or epinephrine (PFA-100^®^-EPI). This membrane has a hole through which anti-coagulated blood passes. These agonists produce platelet adhesion, activation and aggregation, and the formation of a stable platelet plug that occludes the hole. The time necessary to occlude the hole in seconds is the closure time (CT), and it is in reverse proportional to the functional capacity of platelets. 

For the PFA-100^®^ analysis, blood samples were taken into 4.5 mL tubes that comprised a 3.8% solution of sodium citrate as an anticoagulant (BD Vacutainer Becton, Dickinson and Company, Franklin Lakes, NJ 07417-1880, USA). The samples were transported into the reservoir of the non-refundable test cartridges (PFA-100^®^-ADP, and PFA-100^®^-EPI) already incorporated in the device, and both CT were recorded. Without an exemption, both testing analyses were completed between 15 and 60 min after blood samples were obtained.

We established as CT normal reference values for PFA-100^®^ the values obtained from a blood sample of 80 healthy voluntaries in a previous non-published study performed in our hospital. The normal CT ranged from 58 s to 123 s for the PFA-100^®^-ADP and 72 s to 191 s for the PFA-100^®^-EPI. 

Patients with functional platelet counts below 80 × 10^9^/L measured by Plateletworks^®^ were re-checked every 24 h during three consecutive days until they reached the number of platelets needed for surgery.

### 2.3. Statistical Analysis

In the analysis, we only considered the first determination of platelet function for all patients. We established when platelet count was >80 × 10^9^/L as a normal result for Plateletworks^®^. For comparing Plateletworks^®^ and PFA-100^®^ results, we categorized continuous data of PFA-100^®^ in normal or altered, establishing a normal boundary of 58 s to 123 s for the PFA-100^®^-ADP, and 72 s to 191 s for the PFA-100^®^-EPI; consequently, any value out of this range was rated as pathologic. 

The correlation between these categorical variables of Plateletworks^®^ and PFA-100^®^ was performed calculating Cohen’s Kappa coefficient. The interpretation of its values was the following: below 0.2 showed no agreement or poor agreement, 0.21 to 0.4 showed fair or weak agreement, 0.41 to 0.6 showed moderate agreement, 0.61 to 0.80 showed substantial agreement, and 0.81 to 0.99 showed almost perfect agreement [10]. We used the SPSS statistical software program (v. 24) for the statistical analysis.

## 3. Results

We screened 575 patients undergoing surgery due to proximal femoral fracture; 548 patients were excluded, and 27 were randomized to the experimental group or the control group. Seven randomized patients were excluded from the concordance sub-study because PFA-100^®^ was not performed (Figure 1). 

Eventually, we included 20 patients in the sub-study, 7 (35%) men and 13 (65%) women; their median age was 89 years (range from 70 to 98). Sixteen patients were under treatment with 75 mg/day of clopidogrel, three with > 300 mg/day of acetylsalicylic acid (ASA), and only one was in treatment with both antiplatelet agents but with ASA at a dose of 100 mg/day.

Median hemoglobin was 10.8 g/dL (range from 9.5 g/dL to 14.2 g/dL). Median hematocrit was 32.5% (range from 30% to 42%). Fourteen (70%) out of 20 patients had a hematocrit lower than 35% and only six (30%) patients higher than 35%. The median platelet count was 195 × 10^9^/L (range from 89 × 10^9^/L to 461 × 10^9^/L). Five patients had a platelet count below 150 × 10^9^/L and four of those five had a hematocrit lower than 35%. (Table 1).

### Agreement between PFA-100^®^ and Plateletworks^®^ Tests

The median time from blood extraction to analysis was 6 min (range from 4 min to 11 min). The median number of active platelets by Plateletworks^®^ test was 93 *×* 10^9^/L (range from 32 × 10^9^/L to 210 × 10^9^/L). Six (30%) patients had less than 80 × 10^9^/L active platelets. 

The median ADP-Closure Time (ADP-CT) and EPI-Closure Time (EPI-CT) measured by PFA-100^®^ were 107 s (range from 36 s and 300 s) and 184.5 s (range from 72 s to 300 s), respectively. Eight (40%) patients had anormal values of ADP-CT and 10 (50%) patients of EPI-CT.

There was an agreement between PFA-100^®^-ADP and Plateletworks^®^ in 13 patients (eight patients with normal platelet function, and five patients with pathologic platelet function). There was no correlation between the two tests in seven patients (six patients were normal in Plateletworks^®^ and pathologic in PFA-100^®^-ADP, and one was normal in PFA-100^®^-ADP and pathologic in Plateletworks^®^). Th global Cohen’s kappa coefficient was 0.327 (Table 2). 

There was an agreement between PFA-100^®^-EPI and Plateletworks^®^ in 12 patients (eight patients with normal platelet function and four patients with pathologic platelet function). There was no correlation between the two tests in eight patients (six patients were normal in Plateletworks^®^ and pathologic in PFA-100^®^-EPI, and two were normal in PFA-100^®^-EPI and pathologic in Plateletworks^®^). The global Cohen’s kappa coefficient was 0.200 (Table 2).

## 4. Discussion

Our study showed a weak correlation between PFA-100^®^ and Plateletworks^®^ for measuring platelet function. 

One possible explanation is that both methods do not measure the same parameters. While Plateletworks^®^ presents the count of functioning platelets or the count of inhibition based on platelet aggregability, PFA-100^®^ provides information about platelet adhesion. Furthermore, it is uncertain to what extent PFA-100^®^ is useful for monitoring antiplatelet agents in patients with, or at risk for, cardiovascular diseases [6]. Salicylic acid and other non-steroidal anti-inflammatory drugs inhibit platelet function by blocking cyclooxygenase 1 and formation of thromboxane with the elongation of the EPI-Closure Time (CT) in about 95% of healthy people. However, they have a slight to no effect on ADP-CT. Instead, clopidogrel acts over the ADP receptor, and it can prolong ADP-CT in some patients [11]. Some studies show the PFA-100^®^-CT is relatively unresponsive to ticlopidine and clopidogrel [12]. At present, the utility of the PFA-100^®^ in drug monitoring needs to be proven, and therefore, its uses in this indication are limited to research.

Some trials have assessed the feasibility of some point of care tests and have shown a poor correlation between PFA-100^®^ and LTA [13], the method of reference to measure platelet function. However, Plateletworks^®^ have demonstrated a good correlation with LTA for monitoring clopidogrel response [14].

Furthermore, PFA-100^®^ results could be affected by both anemia (<35%) and thrombocytopenia (<150 × 10^9^ platelets) on PFA-100^®^ analysis [15,16]. Both conditions prolong the CT and are common in patients with a proximal femur fracture. In our study, 70% of the patients had anemia and 25% thrombocytopenia (80% of those had also anemia). The high proportion of patients with a hematocrit lower than 35% made a sensitivity analysis impossible by excluding them from the analysis. The median age of the population of our study was 89 years (range from 70 to 98). Therefore, it is highly likely that platelets and hematocrit are altered in a high proportion at this age and more in the context of bone fracture. This could be interpreted as a limitation of PFA-100^®^ for use in clinical practice in this population. 

The number of active platelets threshold measured by Plateletworks^®^ that we established could also affect the correlation between PFA-100^®^ and Plateletworks^®^. It may be argued that a higher functional platelet count threshold may have shown a different concordance between PFA-100^®^ and Plateletworks^®^. However, platelet counts higher than 80 × 10^9^/L have been demonstrated safe to perform surgery under spinal anesthesia [8,9]; therefore, this was the minimal value chosen.

The strengths of our study are that we have not found studies in the literature which compare PFA-100^®^ and Plateletworks^®^. Furthermore, this study was performed in the context of a randomized clinical trial, and the patients were selected prospectively, in the way the data were accurately recollected.

The shortcomings of our study are the small sample size, which is limited to assess the concordance of these tests based on the type of anti-platelet drug used. In addition, PFA-100^®^ was not the method of reference to measure platelet function. Nevertheless, we selected PFA-100^®^ because is a common point-of-care system in many hospitals.

## 5. Conclusions

We have found a weak concordance comparing PFA-100^®^ and Plateletworks^®^. Despite the results, we think that these point-of-care systems are useful and feasible to guide the perioperative management of those patients in treatment with chronic antiplatelet therapy. Further clinical trials are needed to support their applicability in clinical practice.

## Figures and Tables

**Figure 1 jcm-10-00255-f001:**
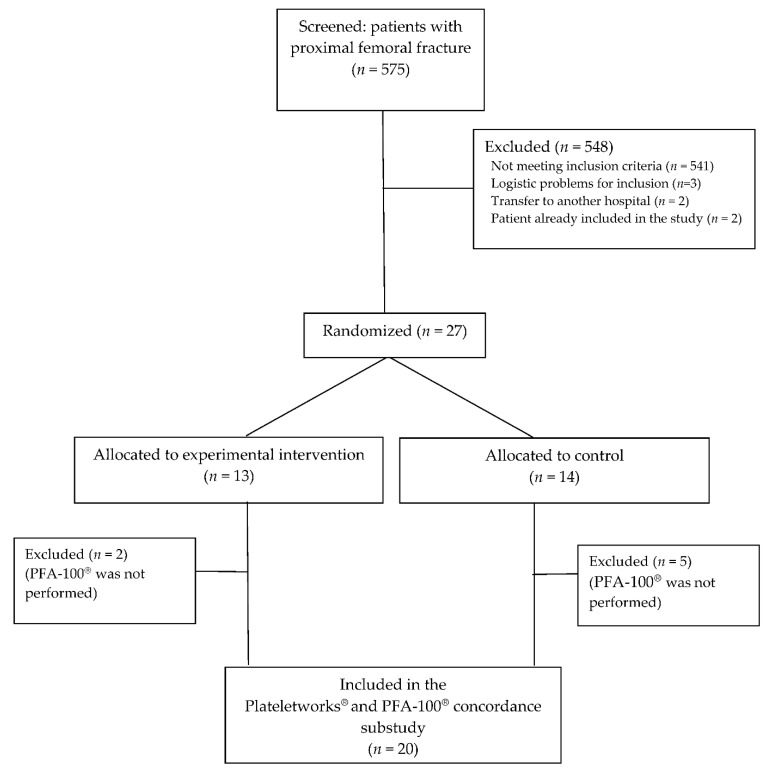
Flowchart of patients. Note: The randomization of patients for the main study was stratified by the center. This flowchart corresponds to the center that participated in the concordance sub-study until the number of patients was achieved.

**Table 1 jcm-10-00255-t001:** Characteristics of included patients and test results.

Case	Age	Sex	Allocated Group	Antiplatelet Drug	Hb at Admission (g/dL)	Hct at Admission (%)	Time from Blood Extraction to Analysis (min)	Total Platelet Count (×10^9^/L)	Active Platelets Count with PLATELETWORK^®^ (×10^9^/L)	Closure Time with PFA-100^®^-ADP (s)	Closure Time with PFA-100^®^EPI (s)
1	89	Woman	Experimental	Clopidogrel	10.9	34	9	101	71	88	251
2	82	Man	Control	Clopidogrel	13.3	39	7	92	74	249	299
3	90	Woman	Experimental	AAS 300 mg	13.6	41	6	461	53	130	300
4	81	Woman	Experimental	Clopidogrel	11.9	33	6	193	50	300	154
5	81	Man	Control	AAS 300 mg and Clopidogrel	9.9	30	6	197	88	147	144
6	89	Woman	Control	Clopidogrel	10.1	32	7	146	108	105	285
7	97	Man	Experimental	Clopidogrel	10.3	30	6	151	126	109	92
8	91	Woman	Experimental	AAS 300 mg	10.3	31	5	212	193	80	143
9	86	Woman	Experimental	Clopidogrel	9.5	30	6	269	123	90	119
10	91	Woman	Control	Clopidogrel	9.7	31	4	274	210	126	300
11	89	Woman	Experimental	Clopidogrel	11.7	32	8	147	89	124	247
12	91	Woman	Experimental	Clopidogrel	12.4	37	11	155	92	97	256
13	77	Man	Control	AAS 300 mg	10.0	30	6	261	166	75	86
14	83	Woman	Control	Clopidogrel	10.2	33	4	198	159	57	94
15	91	Woman	Control	Clopidogrel	12.0	36	5	168	142	79	196
16	90	Woman	Experimental	Clopidogrel	11.2	33	4	315	94	69	173
17	80	Man	Experimental	Clopidogrel	14.2	42	6	187	32	126	300
18	88	Woman	Control	Clopidogrel	10.4	31	6	207	85	36	72
19	70	Man	Experimental	Clopidogrel	12.9	38	6	249	118	300	254
20	88	Man	Control	Clopidogrel	10.8	32	5	89	73	239	155

Hb: haemoglobin; Hct: hematocrit.

**Table 2 jcm-10-00255-t002:** Concordance between PlateletWork^®^ (Helena Laboratories, Beaumont, TX, USA) and PFA-100^®^ (Siemens Canada, Mississauga, ON, Canada) considering the first blood extraction.

		PFA-100^®^ ADP			PFA-100^®^ EPI		
		Normal	Pathologic	Kappa Coefficient	Normal	Pathologic	Kappa Coefficient
PlateletWork^®^	Normal	8	6	0.327	8	6	0.200
	Pathologic	1	5		2	4	

## Data Availability

The data presented in this study are available on request from the corresponding author.

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
