# Peer review of "Correlation between PlateletWorks® and PFA-100® for Measuring Platelet Function before Urgent Surgery in Patients with Chronic Antiplatelet Therapy"

_jcm, 2021, doi:10.3390/jcm10020255_

Round 1
Reviewer 1 Report
In nature of pilot study, I thought this study had worth to be published for further another clinical investigation. As the authors have already mentioned in their manuscript, small number of participants could be major limitation and this might cause type II error in this analysis.
I hope the author will do more research on this field for better understanding.
My only suggestion is that the author are encouraged to use figure when they explain their main result. For example, patient flow chart and the correlation of the PlateletWork should be explained using appropriate figure.
Author Response
Point 1: In nature of pilot study, I thought this study had worth to be published for further another clinical investigation. As the authors have already mentioned in their manuscript, small number of participants could be major limitation, and this might cause type II error in this analysis.
I hope the author will do more research on this field for better understanding.
Response 1: When we designed the study the number of patients was higher, but by restriction in the funding we decided to limit the number of patients for this substudy. This was justified because other studies with a similar number of patients obtained valid results. However, in view of the data, we agree that a higher number of patients had reduced the variability in the results obtained and was increased the certainties of them.
Point 2: My only suggestion is that the author are encouraged to use figure when they explain their main result. For example, patient flow chart and the correlation of the PlateletWork should be explained using appropriate figure.
Response 2: We have included the flow-chart of the study.
We calculated the correlation of the PlateletWork with the PFA100 with Cohen's Kappa to assess the agreement by a categorical variable (normal versus altered result). This limits the possibility to express it as a Figure.
Reviewer 2 Report
The manuscript reports the comparison of two point-of-care platelet function devices: Platelet Function Analyzer PFA-100 and Plateletworks in the context of the measuring platelet function before urgent surgery in patients with chronic antiplatelet therapy. This study is an ongoing, multicenter clinical trial (NCT03231787).
The aim of this manuscript is interesting and important in clinical practice, however I suggest considering its publication rather as a Short communication than a Regular article. The study has many limitations and the Authors described the limitations in the Discussion section. The major disadvantage is a small sample size (20 patients), additionally in this group 5 patients (25%) had a platelet count below 150 x 109/L or 14 patients (70%) had haematocrit lower than 35%. Platelet count (below 150 x 109/L) as well as haematocrit (<30%) may significantly influence the results from PFA-100 measuring. In my opinion, the analysis of the correlation between the results from PFA-100 and Plateletworks should be calculated based on data omitting potential artefacts. Also, the number of patients in the participant group should be large. The pros and cons of current methods of measuring platelet activity, including both PFA-100 and Plateletworks are reported in scientific literature.
The Authors should describe in more details the choosing of closure time normal reference values for PFA-100.
The description of Plateletworks and PFA-100 methods is detailed, an information about working principle can be removed (the paragraph 2.2. Platelet function tests).
Author Response
Point 1:
The manuscript reports the comparison of two point-of-care platelet function devices: Platelet Function Analyzer PFA-100 and Plateletworks in the context of the measuring platelet function before urgent surgery in patients with chronic antiplatelet therapy. This study is an ongoing, multicenter clinical trial (NCT03231787).
The aim of this manuscript is interesting and important in clinical practice; however, I suggest considering its publication rather as a Short communication than a Regular article.
Response 1:
In our opinion, this manuscript has enough information and extension for considering it as a regular article.
Point 2:
The study has many limitations and the Authors described the limitations in the Discussion section. The major disadvantage is a small sample size (20 patients), additionally in this group 5 patients (25%) had a platelet count below 150 x 109/L or 14 patients (70%) had haematocrit lower than 35%. Platelet count (below 150 x 109/L) as well as haematocrit (<30%) may significantly influence the results from PFA-100 measuring. In my opinion, the analysis of the correlation between the results from PFA-100 and Plateletworks should be calculated based on data omitting potential artefacts.
Response 2:
This is a good observation. “…the results from PFA-100 and Plateletworks should be calculated based on data omitting potential artefacts”. However, the high proportion of patients with a hematocrit lower than 35% impossibilities a sensitivity analysis excluding them from the analysis. The median age of the population of our study was 89 years (range from 70 to 98). It is highly likely that platelets and hematocrit are altered in a high proportion at this age and more in the context of bone fracture. This could be interpreted as a limitation of PFA100 for use in clinical practice in this population. We have added this comment to the discussion.
Point 3:
Also, the number of patients in the participant group should be large.
Response 3:
When we designed the study the number of patients was higher, but by restriction in the funding, we decided to limit the number of patients for this substudy. This was justified because other studies with a similar number of patients obtained valid results. However, in view of the data, we agree that a higher number of patients had reduced the variability in the results obtained and was increased the certainties of them.
Point 4:
The pros and cons of current methods of measuring platelet activity, including both PFA-100 and Plateletworks are reported in the scientific literature.
Response 4:
We agree, but there are no studies with a direct comparison between PFA-100 and Plateletworks including the same population. This is the novelty of our study.
Point 5:
The Authors should describe in more detail the choosing of closure time normal reference values for PFA-100.
Response 5:
We have added in the manuscript more details about how we obtained closure time normal reference values for PFA-100.
Point 6:
The description of Plateletworks and PFA-100 methods is detailed, information about working principle can be removed (the paragraph 2.2. Platelet function tests).
Response 6:
In our opinion, it can be useful to mention the working principle of these tests for the readers of the journal.
Round 2
Reviewer 2 Report
Thank the Authors for the responses to my comments from Report 1.
Major comments:
I suggest adding information about the age of study participants (range from 70 to 98) into the Conclusions and the aim of the study (also into the Abstract).
Page 3, line 136
“Twelve randomized 134 patients were excluded from the concordance substudy because PFA-100 was not performed 135 (Figure 1).” In the manuscript there are any figures, there are only tables.
Author Response
Point 1:
I suggest adding information about the age of study participants (range from 70 to 98) into the Conclusions and the aim of the study (also into the Abstract).
Response 1:
In the abstract, we have added the age of the study participants in the results section and in the conclusions, we have included a comment related to the influence of the advanced age in the concordance results.
Point 2:
Page 3, line 136
“Twelve randomized patients were excluded from the concordance substudy because PFA-100â was not performed (Figure 1).” In the manuscript there are no figures, there are only tables.
Response 2:
We submitted Figure 1 independently of the manuscript following the instructions of the journal. Now, we added Figure 1 in this answer. Reviewing Figure 1 we appreciate a mistake in the numbers that have been corrected.
